# *SiMYB19* from Foxtail Millet (*Setaria italica*) Confers Transgenic Rice Tolerance to High Salt Stress in the Field

**DOI:** 10.3390/ijms23020756

**Published:** 2022-01-11

**Authors:** Chengjie Xu, Mingzhao Luo, Xianjun Sun, Jiji Yan, Huawei Shi, Huishu Yan, Rongyue Yan, Shuguang Wang, Wensi Tang, Yongbin Zhou, Chunxiao Wang, Zhaoshi Xu, Jun Chen, Youzhi Ma, Qiyan Jiang, Ming Chen, Daizhen Sun

**Affiliations:** 1Key Laboratory of Sustainable Dryland Agriculture, College of Agriculture, Shanxi Agricultural University, Jinzhong 030801, China; xu1253824860@163.com (C.X.); mrshihuawei@163.com (H.S.); yanhs111@163.com (H.Y.); 18503483681@163.com (R.Y.); wsg6162@126.com (S.W.); 2Institute of Crop Sciences, National Key Facility for Crop Gene Resources and Genetic Improvement, Key Laboratory of Biology and Genetic Improvement of Triticeae Crops, Chinese Academy of Agricultural Sciences (CAAS), Beijing 100081, China; caasluomingzhao@163.com (M.L.); sunxianjun@caas.cn (X.S.); tang_wensi@yeah.net (W.T.); houyongbin@caas.cn (Y.Z.); wangchunxiao@caas.cn (C.W.); xuzhaoshi@caas.cn (Z.X.); chenjun01@caas.cn (J.C.); mayouzhi@caas.cn (Y.M.); jiangqiyan@caas.cn (Q.J.); 3College of Life Sciences, Shanxi Normal University, Taiyuan 030006, China; jijiyan19970316@163.com

**Keywords:** ABA pathway, foxtail millet, high salt stress, MYB transcription factor

## Abstract

Salt stress is a major threat to crop quality and yield. Most experiments on salt stress-related genes have been conducted at the laboratory or greenhouse scale. Consequently, there is a lack of research demonstrating the merit of exploring these genes in field crops. Here, we found that the R2R3-MYB transcription factor *SiMYB19* from foxtail millet is expressed mainly in the roots and is induced by various abiotic stressors such as salt, drought, low nitrogen, and abscisic acid. *SiMYB19* is tentatively localized to the nucleus and activates transcription. It enhances salt tolerance in transgenic rice at the germination and seedling stages. *SiMYB19* overexpression increased shoot height, grain yield, and salt tolerance in field- and salt pond-grown transgenic rice. *SiMYB19* overexpression promotes abscisic acid (ABA) accumulation in transgenic rice and upregulates the ABA synthesis gene *OsNCED3* and the ABA signal transduction pathway-related genes *OsPK1* and *OsABF2*. Thus, *SiMYB19* improves salt tolerance in transgenic rice by regulating ABA synthesis and signal transduction. Using rice heterologous expression analysis, the present study introduced a novel candidate gene for improving salt tolerance and increasing yield in crops grown in saline-alkali soil.

## 1. Introduction

Salt stress adversely affects plant growth and development and has a serious impact on crop yield and quality. Eight hundred million hectares of soil worldwide are affected by salinity according to FAO data. Soil salinity affects over 20% of all arable land globally. This rate is expected to continue to increase [1]. Therefore, the selection of salt-tolerant crops is vital to the maintenance of grain production in saline-alkali soil, the expansion of cultivable land, and the assurance of food security. Foxtail millet (*Setaria italica* L.) belongs to the Gramineae family. It originated in China and has strong natural abiotic stress resistance [2], a small genome, and a short growth cycle. Therefore, it is an ideal model crop to study abiotic stress resistance in gramineous crops [2]. However, very little research has been conducted to date on the functional genome of foxtail millet. Furthermore, its stress-related regulatory network is poorly understood. Identification of the key stress resistance genes in foxtail millet and especially those with important practical field application may help facilitate the improvement of stress resistance in this plant and the other gramineous crops.

MYB-like transcription factors perform various functions during plant growth and development. They regulate anthocyanidin biosynthesis and accumulation, lateral root and pollen development, and phytohormones [3]. *Arabidopsis* induces the expression of *AtMYB2* and *AtMYB15* by regulating ABA content under drought and high salt conditions [4,5]. *GAMYB* in rice is also an important regulator of gibberellin signal transduction [6]. The foregoing genes also play important roles in stress adaptation. The rice R2R3-MYB transcription factor (TF) *OsMYB2* is strongly induced in response to salt and cold stress. *OsMYB2* overexpression enhances tolerance to different abiotic stressors in *OsMYB2* transgenic plants [7]. In *Arabidopsis*, certain R2R3-MYB TFs such as *AtMYB2*, *AtMYB20*, *AtMYB44*, *AtMYB73*, and *AtMYB74* are induced by salt stress and regulate salt tolerance [8]. *ZmMYB3R* is a positive regulator of salt and drought resistance. Its ectopic expression significantly enhances transgenic plant tolerance to drought and salt stress [9]. The alfalfa TF *MYB4* is activated by DNA methylation and/or histone modification in response to salt stress [10]. The poplar MYB TF *PtrSSR1* enhances salt stress tolerance in transgenic plants [11]. Apple *MdMYB88* and its homolog *MdMYB124* regulate root xylem development and cell wall cellulose accumulation under drought conditions. Therefore, these genes regulate water transport under drought stress [12]. Several studies have reported on the roles of MYB TFs in abiotic stress response. However, it is difficult to regulate the strength of abiotic stressors such as salinity because of the long duration of field trials. Hence, most of the foregoing data were obtained from laboratory or greenhouse experiments. There is a lack of field trial data demonstrating the influences of MYB-like TFs on abiotic stress resistance in field crops. Functional evaluation of stress-related genes in field crops may help establish whether they merit further investigation and should be applied in practical breeding research.

In our previous work, we analyzed the transcriptome of foxtail millet subjected to drought and salt stress and found that several MYB TFs responded to these and other stressors [13]. *SiMYB19* and *SiMYB56* were differentially expressed in drought stress transcriptome sequencing and were upregulated in prior studies. In the present study, we found that the MYB TF *SiMYB19* was induced by salt stress. We transformed it into wild type (WT) Kitaake (Ki) rice and obtained stable T3 generation transgenic rice by *Agrobacterium* transformation. We transplanted the transgenic rice lines with the highest salt tolerance from the laboratory to the field and exposed them to various salt concentrations. *SiMYB19* overexpression significantly improved survival in transgenic rice exposed to high salt concentrations (0.5% (*w*/*v*) NaCl) and augmented grain yield in transgenic rice subjected to moderate salt concentrations (0.3% (*w*/*v*) NaCl). Hence, *SiMYB19* could be exploited to improve crop salt tolerance. We also identified the downstream network of *SiMYB19* via qRT-PCR and analyzed the phenotypes of the transgenic rice plants.

## 2. Results

### 2.1. SiMYB19 Gene Structure and Phylogenetic Tree Analysis

In previous research, we performed RNA-seq on a stress-tolerant foxtail millet subjected to drought or salt [13,14] and discovered that the MYB-like TF *SiMYB19* (Seita.1G250600.1) significantly responded to salt stress. *SiMYB19* was localized to the first chromosome of the foxtail millet genome. The genome sequence length was 1562 bp and included three exons, two introns, and an 888 bp long open reading frame (ORF). *SiMYB19* encodes a protein that was 295 amino acids (aa) in length (Figure 1A). The predicted molecular weight (MW) of SiMYB19 was 31.72 kDa, and its isoelectric point was 5.83. SiMYB19 contains two SANT conserved domains and belongs to the R2R3 subgroup of the MYB TF family (Figure 1B,C). The phylogenetic tree showed that the SiMYB19 protein has the highest homology with maize ZmLAF1 and with *Arabidopsis* AtMYB19, AtMYB18, and AtMYB45 (Figure 1C) [15].

### 2.2. SiMYB19 Expression Profile in Foxtail Millet under Different Treatments

We analyzed the *SiMYB19* expression profile after treating seedlings with 100 mM NaCl, 10% (*w*/*v*) PEG6000, 100 μM ABA, 0.2 mM NO_3_^−^ (LN (low nitrogen); nitrogen deficiency) and 100 μM IAA. *SiMYB19* was expressed mainly in the roots (Figure 2A). It was strongly induced by NaCl, drought, LN, and ABA exposures and weakly induced by IAA (Figure 2B–F). Under 100 mM NaCl, *SiMYB19* expression reached a maximum at 6 h and was sixfold higher than that at 0 h (Figure 2B). Under 10% (*w*/*v*) PEG6000, *SiMYB19* expression reached a maximum at 6 h and was twentyfold higher than that at 0 h (Figure 2C). Under LN, *SiMYB19* expression reached a maximum at 24 h (Figure 2D). Under 100 μM ABA, *SiMYB19* expression reached a maximum at 12 h and was sevenfold higher than that at 0 h (Figure 2F).

### 2.3. SiMYB19 Tentatively Localizes to The Nucleus and Activates Transcription

We generated a *SiMYB19-GFP* fusion vector controlled by the CaMV35S promoter to determine subcellular SiMYB19 protein localization. The vectors were transiently transformed in foxtail millet mesophyll protoplasts. The *16318hGFP* vector served as a positive control. The subcellular analysis indicated that the GFP-SiMYB19 protein tentatively localized to the nucleus (Figure 3A). We transformed the pBD-SiMYB19 vector and the negative control pGBKT7 empty vector into the AH109 yeast strain to identify transcription activation mediated by SiMYB19. The pBD-SiMYB19 transformants (BD-SiMYB19) grew on SD/-Trp/-His/-Ade selection medium and SD/-Trp/-His/-Ade plus X-α-gal. By contrast, the negative control pGBKT7 empty vector (BD) did not grow on the selection medium (Figure 3B). Thus, SiMYB19 activated transcription in yeast.

### 2.4. SiMYB19 Overexpression Enhanced Transgenic Rice Seedling Tolerance to Salt Stress

Foxtail millet transformation is difficult to perform. To identify the functions of *SiMYB19,* we transformed it into rice. We identified T3 generation transgenic rice plants by PCR (Appendix A). The qRT-PCR disclosed that in response to 100 mM NaCl, the transgenic rice lines OE-6, OE-8, and OE-9 had higher *SiMYB19* expression levels than the WT (Appendix A). We analyzed greenhouse-raised, PCR-positive transgenic lines for salt tolerance. Under normal growth conditions, the phenotypes of the transgenic and WT rice lines did not significantly differ from each other (Figure 4A). After 100 mM NaCl treatment for 5 d, there was significantly less leaf curling in OE-6, OE-8, and OE-9 than the WT (Figure 4B). The greenhouse salt tolerance analysis revealed that the transgenic line OE-6 had the highest survival rate (Figure 4C). Thus, it was selected for the subsequent salt tolerance analyses in the salt ponds and the field. After recovery to normal growth conditions, the transgenic line survival rates (20–40%) were significantly (*p* < 0.05) higher than that of the WT (5%) (Figure 4A–D). After salt treatment, the malondialdehyde (MDA) concentrations in the transgenic rice were significantly (*p* < 0.05) lower than that of the WT (Figure 4E). Plant height (*p* < 0.05), root length (*p* < 0.05), aboveground fresh weight (*p* < 0.05), underground fresh weight (*p* < 0.01), aboveground dry weight (*p* < 0.01), and underground dry weight (*p* < 0.01) were significantly higher for the transgenic rice plants than the WT (Figure 4F–K). Therefore, *SiMYB19* overexpression significantly improved greenhouse-grown transgenic rice plant salt stress tolerance.

### 2.5. SiMYB19 Overexpression Increased the Yield of Field-Grown Transgenic Rice Subjected to High Salt Stress

We selected the optimal transgenic line OE-6 and assessed its salt tolerance in the field. We set up the control (CK; water), 0.3% (*w*/*v*) NaCl, and 0.5% (*w*/*v*) NaCl treatments. For the control, OE-6 was taller than the WT (Figure 5A). Under 0.3% and 0.5% (*w*/*v*) NaCl, OE-6 was taller and healthier than the WT. Moreover, the differences in health status between OE-6 and WT increased with salt concentration (Figure 5A). Under 0.3% (*w*/*v*) NaCl, the plant height and grain weight per plant were significantly (*p* < 0.01) higher for OE-6 than the WT (Figure 5B,C). Under 0.3% (*w*/*v*) NaCl, the salt tolerance indices of the WT and OE-6 were 98% and 60%, respectively, and the difference was significant (*p* < 0.01) (Figure 5D). Both the salt tolerance and relative salt tolerance indices decreased with increasing salt tolerance (Figure 5D). The relative salt tolerance index of OE-6 was significantly lower than that of the WT (*p* < 0.01) (Figure 5E). Thus, OE-6 exhibited a higher salt tolerance than the WT in the field. In the salt ponds containing 0.3% (*w*/*v*) NaCl, OE-6 grew better and had higher salt tolerance than the WT (Figure 5F).

### 2.6. Salt Tolerance, ABA, and Drought Stress Response in OE-6

A salt tolerance analysis showed that the germination rate of OE-6 was higher than that of the WT under 100 mM, 200 mM, and 250 mM NaCl (Figure 6A,C–F). After 7 d, the final germination rate of OE-6 was significantly higher than that of the WT (Appendix A). Thus, OE-6 had stronger salt tolerance than the WT during germination. ABA is the main stress-related phytohormone [16]. We analyzed the sensitivity of transgenic rice to ABA treatment using tungstate, an ABA synthesis inhibitor. The germination rate of OE-6 was lower than that of the WT under normal conditions (Figure 6B,C) and the ABA content of OE-6 was higher than that of the WT (Figure 6H). Under 5 mM tungstate, however, the seed germination rates (Figure 6B,G) and the ABA levels (Figure 6H) of both OE-6 and the WT were similar. Therefore, *SiMYB19* transgenic rice had relatively greater sensitivity to ABA treatment and *SiMYB19* is implicated in ABA signaling.

Based on *SiMYB19* upregulation induced by PEG in the early growth stages (Figure 2C), we conducted another soil drought experiment. After 15 d severe water restriction followed by 5 d irrigation, OE-6 showed better growth (Appendix A) and significantly increased survival rate (Appendix A) compared with the WT. Hence, OE-6 had a certain degree of drought tolerance.

### 2.7. Expression Analysis of ABA- and Salt Stress-Related Genes in SiMYB19 Transgenic Rice

We used qRT-PCR to analyze the expression of the gene that synthesizes ABA in *SiMYB19* transgenic rice and determine whether salt tolerance is related to ABA pathway gene expression. Under both the control and salt treatments, expression of the ABA synthesis gene *OsNCED3* (9-*cis*-cyclocarotenoid dioxygenase) was higher in the transgenic plants than the WT (Figure 7A). The expression levels of the ABA signal transduction genes *OsPK1* (pyruvate kinase 1) and *OsABF2* (ABRE-binding factor 2) were higher in OE-6, OE-8, and OE-9 than the WT (Figure 7B,C). LEA (late embryogenesis abundant proteins) improve plant salt tolerance and drought resistance [17]. Here, *OsLEA7* expression was higher in the transgenic rice lines than the WT (Figure 7D).

## 3. Discussion

### 3.1. SiMYB19 Is a Positive Regulator That Modulates Field Crop Salt Stress Tolerance

The present study demonstrated that *SiMYB19* overexpression significantly improves salt tolerance in transgenic rice grown in the greenhouse and field. Salinization seriously affects crop growth, yield, and total agricultural production [18,19]. The current global area of salinized land is ~954 million hm^2^, accounts for 7% of the total land area worldwide, and is distributed mainly in Africa, western North America, and Eurasia [20]. In China, the area of saline-alkali land is ~36.66 million hm^2^ [21], and most of it has neither been developed nor utilized [22]. Moreover, it is constantly expanding, severely reduces crop growth and grain yield, and threatens the environment and food security. Salt stress may impede crop growth at different developmental stages. The plant growth cycle cannot proceed normally when the soil salt concentration is >200 mM [23,24]. Here, we showed that *SiMYB19* overexpression increased salt tolerance in field-grown transgenic rice subjected to 0.3% (*w*/*v*) and 0.5% (*w*/*v*) NaCl. Under these conditions, the transgenic rice could grow and develop normally and had significantly superior grain yield and salt tolerance compared to the WT plants (Figure 5A–C). *OsMYB91*, *OsMYB2*, and *OsMLD* (with MYB TF domain) genes were overexpressed in rice [7,25,26]. The results showed that MYB TF had a positive regulatory effect on salt stress, and our newly identified transgenic line OE-6 also showed similar salt stress resistance. Therefore, this study may provide germplasm resources for improving crop salt tolerance. *SiMYB19* has potential value in practical cereal crop breeding research. This gene can improve salt tolerance and increase yield in saline-alkali soil. The stable application of *SiMYB19* may be validated by multi-year field experiments. If these trials are successful, then *SiMYB19* could help expand the global range of arable land.

### 3.2. SiMYB19 Confers Salt Stress Tolerance through an Aba-Dependent Pathway

The four categories of MYB family proteins are MYB-related, R2R3-MYB, 3R-MYB (R1R2R3-MYB), and 4R-MYB [27,28]. Abiotic stress can induce MYB-related genes. Overexpression of the latter in transgenic plants can increase drought and salt stress resistance. Several R2R3-like MYB TFs participate in stress tolerance [7,29]. Here, we found that *SiMYB19* belongs to the R2R3-MYB subgroup (Figure 1B). A phylogenetic tree identified genes with the highest homology, namely, *ZmLAF1*, *OsMYB19*, *TaMYB18*, *SiMYB18,* and *AtMYB45* in maize, rice, wheat, foxtail millet, and *Arabidopsis*, respectively. The foregoing genes belong to the R2R3-MYB TF family (Figure 1C). *SoMYB18* is a sugarcane R2R3-MYB TF that improved salt and drought tolerance in tobacco [30]. *MYB15* overexpression improved drought and salt resistance in *Arabidopsis* by enhancing its ABA sensitivity [31]. Therefore, the R2R3-MYB TF *SiMYB19* identified herein may also confer abiotic stress resistance in other plants.

MYB-related genes regulate ABA-related pathways and enable plants to contend with various stressors. ABA reduces water loss from cells in response to osmotic stress [32]. The genes regulating abiotic stress response are involved in both ABA-dependent and ABA-independent signaling pathways [33,34]. In the former, the class A protein phosphatase PP2Cs represses *SnRK2s* [35], and the ABA signaling pathway is closed. Under stress conditions, ABA production is upregulated, and the phytohormone binds the receptor protein PYR/PYL/RCARs [36] to form a receptor complex with PP2Cs. The SnRK2s is released and is automatically phosphorylated and self-activated [37,38]. It then phosphorylates downstream ABA TFs and regulates the expression of ABA-responsive genes. Here, we found that the expression levels of ABA synthesis signal transduction genes such as *NCED3*, *ABF2*, and *PK1* were higher in *SiMYB19* transgenic plants than in the WT (Figure 7A–D). The high expression levels of these genes observed in OE-8 may be explained by the fact that the plants were sampled at the early stages of salt stress treatment. *SiMYB19* confers salt and drought tolerance in field-grown transgenic rice through the ABA pathway. Previous studies reported that *AtMYB44* and *AtMYB96* control plant drought and ABA responses through the ABA-dependent signaling pathway [39,40]. *OsMYB6* overexpression affected neither the growth nor the development of transgenic rice but increased its ABA sensitivity and enhanced its resistance to drought and salt stress [41]. Overexpression of the MYB19-like protein TaMYB19-B from *Triticum aestivum* improved stress tolerance in transgenic *Arabidopsis*. *TaMYB19-B* was induced by both abiotic stress and exogenous ABA treatment [42]. *TaMYB19-B* overexpression upregulated *RD29A*, *RD22*, and *MYB2* in transgenic *Arabidopsis*. *RD29A* plays a role in an ABA-independent pathway while *RD22* and *MYB2* act through an ABA-dependent pathway [43,44]. No detailed functional analysis of *TaMYB19-B*-mediated stress tolerance has been performed, and its specific association with ABA is unknown. The present study revealed that the action of *SiMYB19* in salt stress tolerance is mediated through ABA synthesis and signal transduction. We confirmed that *SiMYB19* is regulated through an ABA-dependent pathway. Nevertheless, the specific mechanism by which *SiMYB19* regulates the ABA pathway merits further investigation.

### 3.3. SiMYB19 Modulates Drought Stress

Drought and salt stress negatively influence plant growth and crop productivity [45]. The improvement of crop drought tolerance is vital to food security [46,47]. Here, we found that 10% (*w*/*v*) PEG also induced *SiMYB19* (Figure 2C). To clarify the regulatory roles of *SiMYB19* on other types of abiotic stress, we conducted a drought tolerant analysis on OE-6 transgenic rice. After 15 d drought stress and 7 d recovery, the survival rate of OE-6 was higher than that of the WT (Appendix A). Therefore, the roles that *SiMYB19* plays in other types of abiotic stress remain to be determined.

## 4. Methods

### 4.1. Plant Materials, Growth Conditions, and Stress Treatments

For the gene expression profile analysis, foxtail millet (Yugu1) was grown in pots for 3 wks in a greenhouse at 60% RH (relative humidity), 14 h light at 21 °C, and 10 h darkness at 24 °C. The plants were then transferred to Hoagland’s nutrient solution for 3 d and subjected to control, salt stress (100 mM NaCl), osmotic stress (10% (*w*/*v*) PEG6000), abscisic acid (100 mM ABA), auxin (100 mM IAA), or low-nitrogen stress (LN; 0.2 mM NO_3_^−^) treatment. Leaves were sampled for the gene expression analysis at 0, 1, 3, 6, 12, 24, and 48 h. The roots, stems, and leaves of the control plants were collected for organ-specific SiMYB19 expression analysis. All samples were stored at −80 °C, and there were three replicates per sample.

For the salt tolerance assay, transgenic rice seeds were germinated in water for 3 d, and the seedlings were transferred to a 96-well plate floating on water in a hydroponic box for 1 wk. The seedlings were then transferred to nutrient solution for 1 wk (Appendix A). For the salt treatment experiment, 2-wk-old seedlings were transferred to nutrient solution + 100 mM NaCl for 5 d and allowed to recover in plain nutrient solution for 7 d. On day 5 of the salt treatment, plant height, root length, aboveground fresh weight, underground fresh weight, aboveground dry weight, and underground dry weight were measured. The malondialdehyde (MDA) content was determined with Keming’s MDA Detection Kit (micromethod; Suzhou Comin Biotechnology, Jiangsu, China), and the experiment was performed in triplicate. Seedling survival was evaluated on day 7 of the recovery culture [48]. For the soil-based drought tolerance experiments, the WT and OE-6 (transgenic rice line) seedlings were transferred to soil. At the six-leaf stage, watering was stopped for 15 d and resumed for another 7 d.

### 4.2. Sequence Alignment and Phylogenic Tree Construction

The exon-intron structure of *SiMYB19* was plotted by the gene structure display server (GSDS) program (http://gsds.cbi.pku.edu.cn/, accessed on 5 October 2021). The MYB-like TF sequences were downloaded from https://blast.ncbi.nlm.nih.gov/Blast.cgi, accessed on 5 October 2021. The SiMYB19 phylogenic tree was constructed by the neighbor-joining (NJ) method via MEGA7 software (https://www.megasoftware.net/dload_win_gui, accessed on 5 October 2021) [49].

### 4.3. Rice Transformation and Subcellular Simyb19 Localization in Foxtail Millet

The *SiMYB19* coding region was amplified by PCR. The PCR product was then digested and ligated into the binary vector pCAMBIA1390 driven by a ubiquitin promoter, and the plasmid pCAMBIA1390-SiMYB19 was obtained. The latter was transformed into *Oryza sativa* cv. Kitaake via *Agrobacterium* [50]. The SiMYB19-16318hGFP fusion protein expression vector was constructed by inserting full-length *SiMYB19* cDNA into the *Bam*HI restriction site of the 16318hGFP vector [51]. The SiMYB19-16318hGFP fusion protein expression vector was transiently transformed via PEG into foxtail millet protoplasts isolated from plants at the two-leaf and one-heart stages. The 16318hGFP empty vector was the negative control. Green fluorescence was observed under a confocal laser scanning microscope (Zeiss LSM980; Carl Zeiss AG, Oberkochen, Germany) at ×40. The image was captured with ZEN software (Carl Zeiss AG). The PCR primers are listed in Appendix A.

### 4.4. Transcription Activation Assay in Yeast

To understand the transcriptional activation of SiMYB19, negative control pGBKT7(BD) [52] and pGBKT7-SiMYB19(BD-SiMYB19) plasmids were transformed into AH109 yeast [53] according to the supplier’s (Clontech Laboratories, Mountain View, CA, USA) instructions. The transformed yeast was spread onto the SD/-Trp solid plate and incubated at 28 °C for 2 d. Single clones on the SD/-Trp solid medium were shaken for 18 h and the yeast cells were diluted 10^0^-, 10^1^-, 10^2^, and 10^3^-fold. The diluted yeast cells were spread onto SD/-Trp and SD/-Trp/-His/-Ade/X-α- gal plates and incubated at 30 °C for 3 d. The transcriptional activation of SiMYB19 was identified based on plant growth status. The primers used are listed in Appendix A.

### 4.5. Salt Tolerance Analyses of SiMYB19 Transgenic Rice in the Field and Salt Ponds

OE-6 was selected for the subsequent salt tolerance analyses in the salt ponds and the field. For the latter, the treatments included the control, 0.3% (*w*/*v*) NaCl, and 0.5% (*w*/*v*) NaCl. The conductivities of the 0.3% (*w*/*v*) and 0.5% (*w*/*v*) NaCl treatments were ~6 ms/cm and ~10 ms/cm, respectively. Salt concentrations were kept constant throughout the growth period by irrigating the substrates with saline solution. The duration and intervals of saline irrigation were based on the electrical conductivity of the field water. For the salt pond experiment, the treatments included the control and 0.3% (*w*/*v*) NaCl. Salt tolerance was assessed mainly by observing and comparing relative plant growth. For the field experiment, the yields of the OE-6 and the WT plants exposed to various salt concentrations were measured at maturity. The salt tolerance index was evaluated as follows:(1)$$\mathrm{Salt}\;\mathrm{tolerance}\;\mathrm{index}=\;(1\times N_{1}+2\times N_{2}+3\times N_{3}+4\times N_{4}+5\times N_{5})/5\times N$$
where N is the total number of plants and N_1_, N_2_, N_3_, N_4_, and N_5_ are the numbers of plants with >75%, 75–50%, 50–25%, <25%, and 0% green leaves, respectively. The relative salt tolerance index is based on the ratio of the transgenic line salt tolerance index to the WT salt tolerance index under the same conditions [54].

### 4.6. Salt Tolerance Analysis of Germinating Transgenic Rice Seeds

Transgenic rice seeds were disinfected with 2.5% (*w*/*v*) sodium hypochlorite solution for 30 min and soaked in 100 mM NaCl, 200 mM NaCl, or 250 mM NaCl. Salt tolerance was evaluated by enumerating the seeds that had germinated under the different salt treatments. To evaluate the ABA sensitivity of *SiMYB19* transgenic rice, the seeds were treated with tungstate, an ABA synthesis inhibitor [55] and the germination rates were calculated every 24 h until all seeds were fully germinated. The experiment was performed in triplicate. The sprouts were then stored at −80 °C and their ABA content was measured with a plant ABA ELISA Kit (JL13378-48T; Jianglai Biological, Beijing, China).

### 4.7. RNA Extraction and qRT-PCRs

The WT and *SiMYB19* transgenic OE-6, OE-8, and OE-9 seedlings subjected to various stressors were used in a gene expression analysis. Total plant RNA was extracted from the seedlings by the TRIzol method with a Zhuangmeng Total RNA Extraction Kit (Zoman Biotechnology Co., Beijing, China). The cDNA was synthesized with a TransScript One-Step gDNA Kit (TransGene, Beijing, China). RT-PCR was performed with a Real Master Mix SYBR Green Kit (TransGene, Beijing, China) using the cDNA as the template. The primers are listed in Additional Appendix A. The qRT-PCR analysis was conducted with a fluorescent quantitative PCR instrument (ABI7500; Applied Biosystems, Foster City, CA, USA). Relative gene expression was calculated by the 2^−^^△△Ct^ method [56] and normalized based on the actin genes in foxtail millet (Si001873m.g) and rice (LOC_Os03g50885).

## 5. Conclusions

*SiMYB19* overexpression plays important roles in improving salt tolerance in field crops. Salt tolerance was significantly improved in transgenic rice lines during the germination and seedling stages. Moreover, their yield and salt tolerance in the field and salt ponds were increased. *SiMYB19* was induced by drought, high salt, low nitrogen, and ABA. Downstream gene expression analysis indicated that *SiMYB19* confers salt tolerance to transgenic field rice through the ABA pathway. Hence, this gene could significantly improve salt tolerance and increase yield in crops grown on saline-alkali land, which, in turn, could expand arable land and maintain food security worldwide.

## Figures and Tables

**Figure 1 ijms-23-00756-f001:**
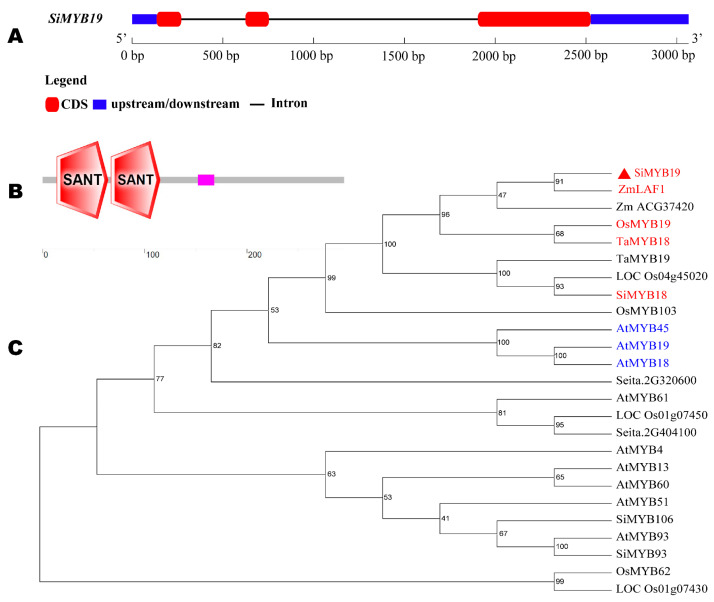
SiMYB19 gene and protein structure and phylogenetic tree. (**A**) SiMYB19 gene structure. Blue is untranslated region (UTR). Red is exon. Black is intron. (**B**) SiMYB19 protein structure. Red polygon is SANT conserved domain. Numbers represent length of aa sequence from *N*-terminal to *C*-terminal. (**C**) SiMYB19 phylogenetic tree. Solid red triangle is SiMYB19. Red indicates highest homology to genes in maize (Zm), rice (Os), wheat (Ta), and foxtail millet (Si). Blue indicates highest homology to genes in *Arabidopsis* (At).

**Figure 2 ijms-23-00756-f002:**
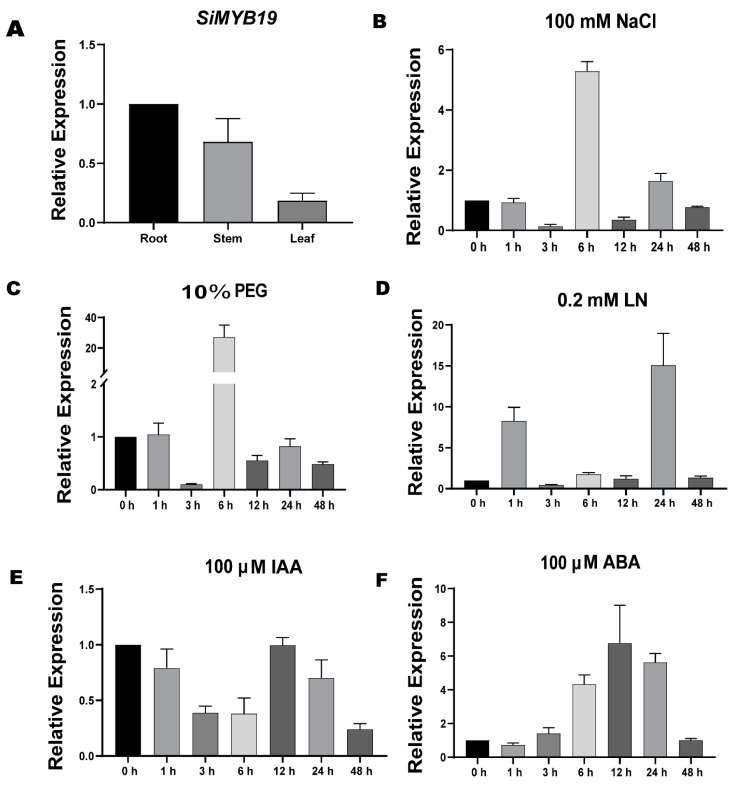
*SiMYB19* expression profile in foxtail millet. (**A**) *SiMYB19* expression in roots, stems, and leaves and in response to (**B**) 80 mM NaCl, (**C**) 10% (*w*/*v*) PEG6000, (**D**) low nitrogen (LN; 0.2 mM NO_3_^−^), (**E**) 100 μM auxin (IAA), and (**F**) 100 μM abscisic acid (ABA). *SiMYB19* expression was determined by qRT-PCR. Data are means ± SD (*n* = 3).

**Figure 3 ijms-23-00756-f003:**
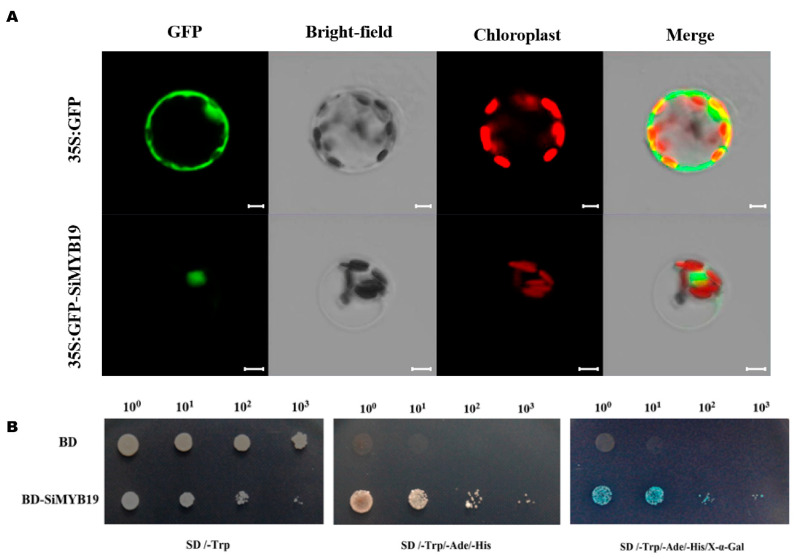
SiMYB19 subcellular localization and transcription activation assay. (**A**) Subcellular SiMYB19 localization in foxtail millet mesophyll protoplasts. The 35S:GFP-SiMYB19 and 35S:GFP control vectors were transiently expressed in foxtail millet mesophyll protoplasts. Fluorescence was observed under a Zeiss LSM980 confocal laser scanning microscope after 16 h transformation. Red is caused by chlorophyll fluorescence in chloroplasts. (**B**) SiMYB19 transcriptional activation assay. pGBKT7-SiMYB19 (BD-SiMYB19) was the expression vector, and the negative control pGBKT7 (BD) was the empty vector. Finally, dilution gradients are represented by 10^0^, 10^1^, 10^2^, and 10^3^.

**Figure 4 ijms-23-00756-f004:**
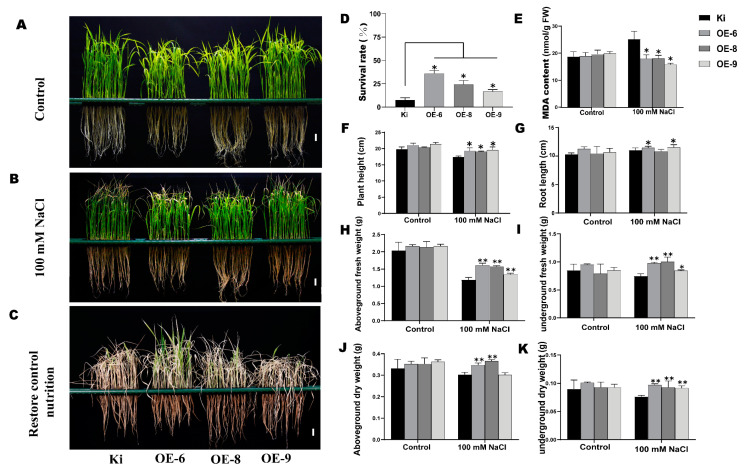
Salt tolerance in greenhouse-raised *SiMYB19* transgenic rice plants. (**A**,**B**) Phenotypes of WT (Ki) and transgenic rice plants under normal and 100 mM NaCl conditions. Bar = 2 cm. (**C**) After 5 d treatment with 100 mM NaCl, all plants were restored to normal conditions for 7 d. Bar = 2 cm. (**D**) Survival rates of WT and transgenic rice plants after restoration to normal conditions. (**E**) Malondialdehyde (MDA) content. (**F**–**K**) Plant height, root length, aboveground fresh weight, underground fresh weight, aboveground dry weight, and underground dry weight under normal growth conditions and 100 mM NaCl treatment. Data are means ± SD (n = 4), * *p* < 0.05, ** *p* < 0.01, *t*-test.

**Figure 5 ijms-23-00756-f005:**
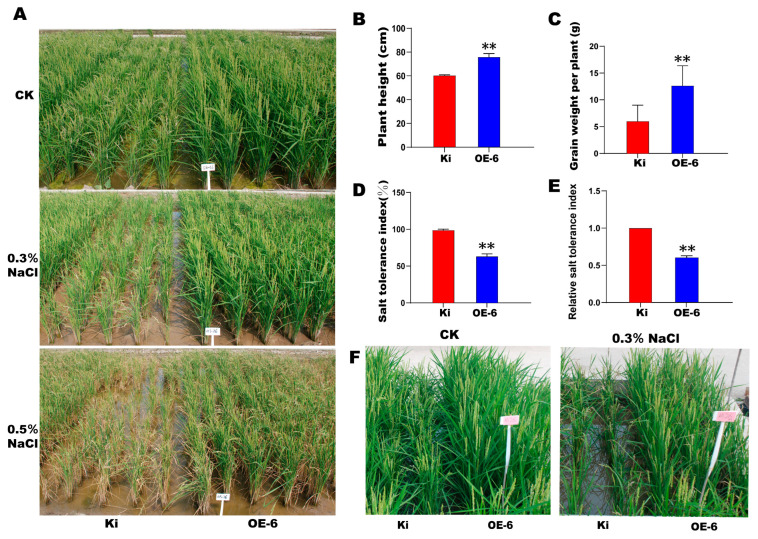
Salt tolerance analysis of *SiMYB19* transgenic rice grown in the field and in salt ponds. (**A**) Phenotypes of transgenic rice line OE-6 and WT (Ki) under normal field conditions (CK), 0.3% (*w*/*v*) NaCl, and 0.5% (*w*/*v*) NaCl. (**B**,**C**) Plant height and yield per WT and OE-6 plant under 0.3% (*w*/*v*) NaCl in the field. (**D**,**E**) Salt tolerance and relative salt tolerance indices of WT and OE-6 in the field. (**F**) Phenotypes of WT and OE-6 under normal conditions and 0.3% (*w*/*v*) NaCl treatment in salt ponds. ** *p* < 0.01.

**Figure 6 ijms-23-00756-f006:**
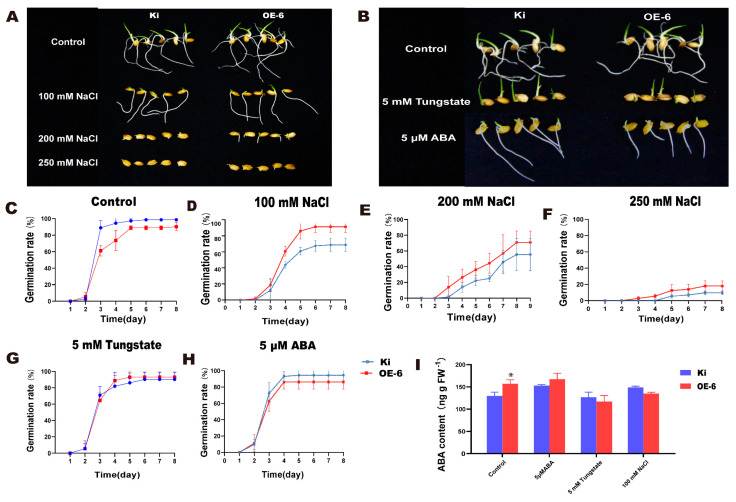
ABA sensitivity analysis of *SiMYB19* transgenic rice. (**A**) Seed germination rates of WT and OE-6 exposed to various NaCl concentrations (*n* = 3; 24 seeds/replicate). (**B**) Seed germination rates of WT and OE-6 under control and 5 mM tungstate treatments. (**C**–**H**) Seed germination rates of WT and OE-6 under control, 100 mM NaCl, 200 mM NaCl, 250 mM NaCl, and 5 mM tungstate treatments. (**I**) ABA levels in OE-6 and WT. Data are means ± SD (*n* = 3), * *p* < 0.05, *t*-test.

**Figure 7 ijms-23-00756-f007:**
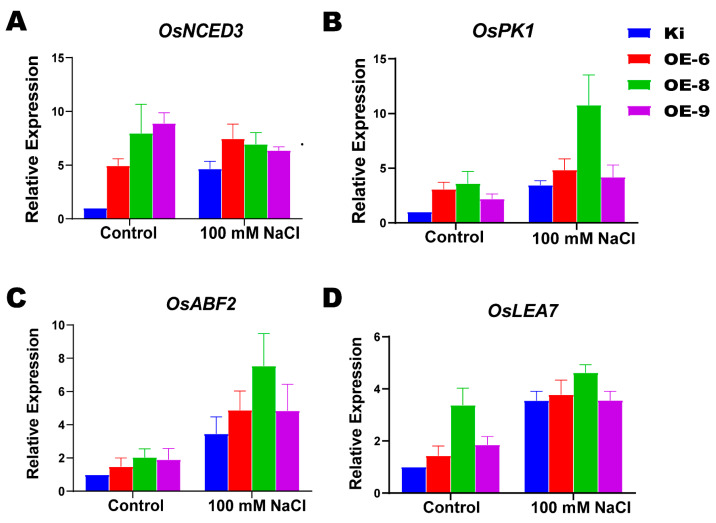
Expression analysis of genes related to ABA synthesis and ABA pathways in *SiMYB19* transgenic and WT plants. (**A**) Relative expression levels of ABA synthesis-related gene *OsNCED3* in WT and transgenic rice. (**B**,**C**) Relative expression levels of ABA signal transduction pathway-related genes *OsPK1* and *OsABF2*. (**D**) Relative expression levels of stress-related gene *OsLEA7*.

## Data Availability

The data that support the findings of this study are available from the corresponding author upon reasonable request.

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
