# Peer review of "SiMYB19 from Foxtail Millet (Setaria italica) Confers Transgenic Rice Tolerance to High Salt Stress in the Field"

_ijms, 2022, doi:10.3390/ijms23020756_

Round 1

Reviewer 1 Report

Foxtail millet (Setaria italica) is attractive to plant scientists as a model plant because of several distinct characteristics, such as its short stature, rapid life cycle, sufficient seed production per plant, self-compatibility, true diploid nature, high photosynthetic efficiency, small genome size, and tolerance to abiotic and biotic stress. Millets are known to possess unique features of resilience to adverse environments, especially infertile soil conditions, although the underlying mechanisms are yet to be determined. In addition, the study on the genetic resources of foxtail millet largely lag behind those of the other model plants such as Arabidopsis, rice and maize.

According to these the aim an topic of the present MS is intersting and up to date.

Recent studies focused on its responses on nitrogen and phosphate limitations, drought stress,  low-light stress. Salt stress induced effect on germination or on growth and photosynthetic characteristics of foxtail millet seedlings have been also published.

The present MS presented the stress related expression of SiMYB19, and the effect of its overexpression.

The experiment is well designed, the results are correct, and was presented demandingly. 

Minor comments and questions:

  1. The authors not enough higlight the previous results based on they chose the  SiMYB 19, this part should be completed in the introduction.
  2. In line 238: "SiMYB19 is invaluable in practical cereal crop breeding
    research. It can improve salt tolerance, increase crop yield in saline-alkali soil." Please re-write this sentence, as it is too
    uplifting.
  3. Please discuss the results of Fig. 2 in order to reveal the putative background of the expression peak in SiMYB19 trancript level after NaCl, PEG or LN treatments.
  4. The conclusion is only a list of the highlights of the results. please re-write is in order to be more synthesised.

Author Response

Dear Reviewer:

Thank you for taking time out of your busy schedule to review the manuscript. Per your guidance, we have carefully corrected and revised the text of the article.

Reviewer 2 Report

In this article, Xu and colleagues describe and molecularly characterize the MYB gene SiMYB19 in foxtail millet and evaluate its potential biotechnological use to generate transgenic rice lines resistant to salt stress in the field and salt ponds. The work is well done, well written and illustrated, but there are certain aspects, mainly related to materials and methods, that need to be addressed and improved.

Introduction:

Line 41. Add family after Gramineae.

Line 51. Specify a little bit more the relationship of MYB-like transcription factors with phytohormones.

Line 65. Change “content” by “intensity” or “strength”.

Results

2.1

In my opinion, the title of the first section of the Results should be changed. First, I can not find any protein sequence alignment in the article or in its supplementary material. Is it missing or authors refer to the phylogenetic anlysis in the form of the phylogenetic tree shown? It would be nice to have this alignment as a supplementary figure. Anyway, authors should consider to renamed this section as “SiMYB19 gene and protein structure, and phylogenetic analysis.”

Figure 1 and legend

In Fig. 1 panel A, figures are not separated of units (in this case pb). This happens here but also in many other cases all over the paper. Authors should take care on this issue.

Indicate in the legend the meaning of the numbers down the protein structure (I gess they are aminoacids from the N- to the C-terminus). Indicate also in the legend, the abbreviations of the different plant species used in the phylogenetic analysis (Sim, Setaria italica, Zm, Zea mays, etc.).

2.2 Add at the end of the title “under different treatments”

The first sentence of this section lists the different treatments for which SiMYB19 expression is tested. However, it does not include IAA, which is then cited. Please, include it, and also specify what LN (low nitrogen) means.

In figure 2, again figures are not separated from units (100mM, or 0h, for example). They must be separated.

2.3

Authors should not ensure that SiMYB19 is localised in the nucleus if they have not shown with DAPI, or similar staining, that the GFP signal colocalizes with DAPI. This is most likely the case, but without the DAPI staining it is better to soften the assertion. Change the title of the section accordingly.

Indicate in figure legend 3, that red color is due to chlorophyll fluorescence. In this regard, I have a small doubt, why are chloroplast indicated in the figure as “Chloroplast II”? Why is this “II” needed for?

2.4

I could not find in materials and methods how the rice transgenics lines used in this article were generated and what are their characteristics (promoter used, cloned region for the overexpression construct, etc.). This is a major drawback since this information should be included in materials and methods, even more so when the levels of overexpression of SiMYB19 in the transgenics are relatively low as shown in figure S2. I also could not find in the supplementary material the sequence of the primers that were used to genotype the transgenic lines.

Figure 1A and 1B, should be Fig. 4A and 4B.

Please, indicate that MDA is Malondialdehyde.

2.5

Figure 5. I think asterisks are missing in the blue bar of Fig. 5E. In the legend, *P<0.05 is not needed if there is no case.

2.7

Gene names should appear in full the first time they are cited (e.g. OSNCED3, OsPK1, etc) in order to make it easier for potential readers to understand the article.

Authors chose overexpression line 6 for their studies because it showed the highest survival rate in their greenhouse salt tolerance analysis. However, the ABA related genes and OsLEA7 are more overexpressed in line 8. Authors should minimally discuss what might account for this apparent paradox.

Discussion

In terms of the biotechnological impact that the transgenic rice lines overexpressing SiMYB19 could have in agronomy, authors should compare in the discussion the resistance level of these lines with those of other transgenic lines overexpressing other MYB genes or any others that confer salt resistance. Are these resistance levels similar, lower or higher than those of the new generated lines?

In my opinion, section 3.3 of the discussion should be included in the results section, as it describes results of a new test to determine the resistance of the overexpression lines to another abiotic stress, drought.

Line 271. Add “from Triticum aestivum” after TaMYB19-B.

Methods

Line 297. Please, do not repeat “h” every time (i.e. 0, 1, 2, 3…and 48 h).

Line 317. “The MYB-like transcription factor (TF) sequence was downloaded from https://blast.ncbi.nlm.nih.gov/Blast.cgi.” I suppose that “sequence” must be plural since they must be those obtained to perform the phylogenetic analysis.

Line 338. Do authors mean “yeast” when they write “plant”?

Line 340 “The greenhouse salt tolerance analysis revealed that the transgenic line OE-6 had the highest survival rate. Thus, it was selected for the subsequent salt tolerance analyses in the salt ponds and the field.” In my opinion, this information must be included in the result section.

Suplementary materials

Figure legend S1 and S2. Indicate what Ki means.

Figure legend S3. Indicate the days of treatment.

Table S1 contains different font types. Please, unify.

Author Response

(The authors gave the same response as above.)

Round 2

Reviewer 2 Report

First of all, I would like to thank the authors for the effort they have made to answer most of my requests, which, in my opinion, improves the article. However, there are still a few that have not been resolved and it would not cost much to do so. These are set out below:

Section 2.2

Line 160. Why the 100 µM IAA is not still included in the sentence listing the different treatments as I asked?

Section 2.3

Please, indicate in figure legend 3, that red color is due to chlorophyll fluorescence of chloroplasts.

Section 2.4

I cannot yet find in the table of supplementary material the sequence of the primers used for genotyping the transgenic lines in figure S1, or are they SiMYB19 F and R? If there were a column in the table specifying the purpose of each primer pair, it would be easier to know.

Section 2.5

The difference in the bars in Fig. 5E appears large enough to be statistically significant. However, there are no asterisks above the blue bar. Please, check as I asked.

Section 2.7

OsPK1: To my knowledge, PK1 is pyruvate kinase 1, not “sphingosine kinase 1” as authors state in the paper. Please, check it, and correct if necessary.

Author Response

We have replied to the reviewers' comments. Please see the attachment
